# Microfluidic Microbial Bioelectrochemical Systems: An Integrated Investigation Platform for a More Fundamental Understanding of Electroactive Bacterial Biofilms

**DOI:** 10.3390/microorganisms8111841

**Published:** 2020-11-23

**Authors:** Stéphane Pinck, Lucila Martínez Ostormujof, Sébastien Teychené, Benjamin Erable

**Affiliations:** Laboratoire de Génie Chimique, Université de Toulouse, CNRS, INPT, UPS, 31432 Toulouse, France; stephane.pinck@ensiacet.fr (S.P.); lucila.marinezostormujof@toulouse-inp.fr (L.M.O.); sebastien.teychene@ensiacet.fr (S.T.)

**Keywords:** microfluidics, microfabrication, bioelectrochemical systems, microbial fuel cell, electroactive biofilms, extracellular electron transfer

## Abstract

It is the ambition of many researchers to finally be able to close in on the fundamental, coupled phenomena that occur during the formation and expression of electrocatalytic activity in electroactive biofilms. It is because of this desire to understand that bioelectrochemical systems (BESs) have been miniaturized into microBES by taking advantage of the worldwide development of microfluidics. Microfluidics tools applied to bioelectrochemistry permit even more fundamental studies of interactions and coupled phenomena occurring at the microscale, thanks, in particular, to the concomitant combination of electroanalysis, spectroscopic analytical techniques and real-time microscopy that is now possible. The analytical microsystem is therefore much better suited to the monitoring, not only of electroactive biofilm formation but also of the expression and disentangling of extracellular electron transfer (EET) catalytic mechanisms. This article reviews the details of the configurations of microfluidic BESs designed for selected objectives and their microfabrication techniques. Because the aim is to manipulate microvolumes and due to the high modularity of the experimental systems, the interfacial conditions between electrodes and electrolytes are perfectly controlled in terms of physicochemistry (pH, nutrients, chemical effectors, etc.) and hydrodynamics (shear, material transport, etc.). Most of the theoretical advances have been obtained thanks to work carried out using models of electroactive bacteria monocultures, mainly to simplify biological investigation systems. However, a huge virgin field of investigation still remains to be explored by taking advantage of the capacities of microfluidic BESs regarding the complexity and interactions of mixed electroactive biofilms.

## 1. Introduction

The ability of bacteria to transfer electrons from or to a solid material (i.e., extracellular electron transfer (EET)) has long been known in the domains of microbial corrosion or biogeochemistry [1,2] but it is only during the last twenty years that it has been used to harvest energy or to catalyze specific electrochemical reactions in bioelectrochemical systems (BESs) [3,4,5,6]. In such engineering systems, bacteria considered as electrochemically active (or electroactive) are able to exchange electrons with electrodes, which they can use as extracellular electron donors or acceptors. The detailed mechanisms allowing the passage of electrons between the bacterial cells and the surface of the solid material of the electrodes are diverse [7] (Figure 1).

EET may occur either by direct physical contact between bacteria and the electrode [11], through redox proteins attached to the bacterial outer membrane [12], or by bacterial nanowires [13,14]. It can also be mediated by electron shuttles [15,16]. Most of these very sophisticated mechanisms are still being investigated by electrochemical, microscopic, genetic and molecular engineering approaches [14,17]. These EET mechanisms are expressed in a very heterogeneous way at the scale of a bacterial consortium, given the species-dependent nature of EET mechanisms. For example, the nanowires and cytochromes of the outer membranes are not universally expressed, synthesized or induced by the same bacterial species [13,14,18,19]. However, the wide variety of micro-organisms within a microbial community permits the cohabitation of various EET strategies that ultimately improve the electrocatalytic activity of bacterial consortia, allowing a wider range of applications of BESs in the fields of bioenergy, biorefineries, biomass, wastewater and organic waste treatment.

Electroactive bacteria (*Geobacter sulfurreducens*, *Shewanella oneidensis*, *Desulfuromonas acetoxidans*, *Pseudomonas aeruginosa*, *Geothrix fermentans*, *Geobacter metallireducens*, *Geobacter bremensis*, *Geoalkalibacter* sp., *Lactococcus lactis*, *Rhodobacter capsulatus*, *Thermincola ferriacetica*, etc.) [20] can collect electrons from a wide range of soluble substrates [3,4] and transfer them to an electrode in the anodic compartment of a BES. In addition, some of the electroactive bacteria (*Geobacter sulfurreducens* again, but also *Clostridium scatologenes*, *Clostridium ljungdahlii*, *Escherichia coli*, *Sporomusa ovata*, etc.) can collect electrons from solid electrodes, i.e., at the biocathodes of microbial fuel cells (MFCs) [21,22], microbial electrosynthesis systems (MESs) [23,24,25,26], microbial electrolysis cells (MECs) [27] or other specific BESs for pollutant remediation [28,29]. Regardless of whether they are anodic or cathodic, electroactive bacteria organize themselves as a biofilm on the surface of the electrode. Their growth dynamics and electrochemical activity on the electrode surface are continuously evolving according to the microbial community populations and the physicochemical conditions of the interface formed between the electrode and the bulk solution; (i) the source of electroactive bacteria (pure culture, synthetic cocktails, consortia), (ii) the properties of the electrode, such as the nature of the material, its topography, its potential, or (iii) the parameters of the bulk, such as temperature, pH, salinity, hydrodynamics and concentrations of soluble nutrients, strongly affect the development of biofilms of electroactive bacteria on the electrodes, and the nature and performance of the EET mechanisms that predominate, mainly within the biofilm. Among the series of parameters described in the literature as having a demonstrated influence on the electrochemical activity of the electrode-electroactive biofilm interfaces (Figure 2), some are fixed and others are modifiable during the formation of electroactive biofilms.

Therefore, the investigation of electroactive biofilms is a multiscale challenge (molecules → cells → biofilm) with a living electrocatalytic interface in evolution as the object of study. However, in the literature, most of the work is conducted at the global scale of a biofilm, i.e., at a rather macroscopic scale of understanding of the phenomena with the implementation of macro electrodes and macro BES. Miniaturizing experimental systems is, however, a very interesting alternative way to study local phenomena at the scale of internal biofilm or even of individual bacterial cells. Thus, the combined or individual use of microelectrodes [30,31] and/or BES microreactors [32,33,34] are dedicated experimental tools giving access to these phenomena of interest at a deeper scale than that of the whole electroactive biofilm. The scale-down of BESs, thanks to microfluidic reactor concepts, allows the experimental systems to be further miniaturized by also ensuring precise control of the electrolyte flow. The latter would be an important parameter because shear stress [35], concentration gradient [36,37] and interaction with planktonic bacteria [35] have an impact on the colonization and formation of biofilm on electrode surfaces.

The idea of using microfluidics to miniaturize BESs was born from their compatibility with easy microfabrication technologies, such as photolithography and soft-photolithography, which are low-cost fabrication methods. Regardless of the manufacturing processes, microfluidic BESs can be classified into two categories. A first category would be the two-electrode set-ups, which are either scaled down macro MFCs with a reactor volume between milliliters and the microlitre [38,39,40,41,42,43] or co-laminar microfluidic MFCs [44,45,46,47,48,49]. A second category would be the three-electrode set-ups that have often been used in more fundamentally oriented studies [50,51,52,53] in reactors where at least one of the dimensions (height, length, or width) is micrometric.

Microfluidic BESs are used as model experimental systems to develop larger scale MFCs, e.g., to test a new electrode material for a microbial fuel cell, or to design proper microfluidic MFCs, whether single [38,39,41,48] or stacked [43] in order to generate energy. Since they make it possible to control the experimental conditions more sensitively at the interface of the electrode and the bulk solution, microfluidic BESs are also used for the rapid detection of electroactive microorganisms [54,55] and as biosensors for environmental monitoring [44,56,57,58,59]. The unique properties of microfluidic systems make them useful in the field of fundamental research, as precise control over the flow would be a critical tool for promoting mass transport or for changing the interfacial conditions at the biofilm–electrolyte interface in electroactive biofilm studies. Microfluidic microbial systems are also described in the study of EET involving nanowires [60] or energy taxy [61], for example.

Some recent reviews [62,63,64,65] have already described the functioning and limitations of microfluidic BESs. However, they have mainly focused on microfluidic MFCs and related energy efficiency issues. Beyond this much investigated field, microfluidic BESs also have many implications and uses in the more fundamental arena of understanding. The present review proposes to focus on the results and fundamental insights that microfluidic BESs have brought to the fundamental understanding of the field of electroactive biofilms. First, the different microfluidic BES systems are presented with their individual limitations and advantages. Then, attention is drawn to the knowledge that microfluidic systems have brought to the field of electroactive biofilms. Finally, perspectives for how microfluidic systems could be used to investigate parameters influencing electroactive bacteria in an electrochemical system are proposed.

## 2. Microfluidic Microbial BES

### 2.1. Scaling Down the BES

Macro sized BESs are well-studied systems [3,9,66,67,68,69,70] but, as they are multiscale devices with biological micro objects as catalysts, they are complex and our understanding of them on the macroscale is hindered by the numerous elements involved. Scaling down the BES reactor volume would ensure a more precise and better-defined understanding of the interaction between them, especially between the electroactive bacteria and the electrodes inside the electrolyte. These systems would be simplified, as it would be easier to obtain accurate control over the parameters and the elements involved. However, just scaling down the dimensions would do little to simplify the system. To gain an accurate and precise understanding of the interaction between electroactive bacteria or biofilms and electrodes, controlling the flow inside the BES would also be necessary, as it is a key parameter of the sheer stress, concentration gradient and interaction between the bacteria and the electrode, which acts as support material for the biofilm formation [43,44,45].

Microfluidic technology contributes greatly to these aspects, as a technology that allows submillimetric working volumes to be used while keeping precise control of the flow [19,48,50,52]. The devices designed with the aim of achieving these parameters are called microfluidic BESs. They are BESs with dimensions around a milliliter and less, where the flow is precisely controlled.

### 2.2. A Wide Range of Designs and Materials

The design and construction techniques used to obtain a microfluidic BES are various [32,38,39,40,41,43,44,45,46,47,48,49,50,51,52,53]. The materials used depend on the technique applied to design the cell. They could be of the same nature as in macroscale BESs, i.e., non-conductive plastic, silicon, glasses, nafion proton exchange membranes [38,39,71,72] in scaled down, milliliter sized BES [42,49,53,73,74,75,76,77].

Most widely used approaches for the fabrication of microfluidic chips rely on soft photolithography techniques and cast molding of a polymer or hot embossing. Molds for microfluidic manufacturing using soft lithography techniques have traditionally been made using SU-8 technology [78] (Figure 3A). This liquid photoresist is spin-coated on a silicon wafer. Then, a series of long processes are combined: baking to dry the deposited layer of liquid photoresist, UV exposure through a mask to print the design patterns on the resin, post-exposure baking, slow cooling to room temperature to avoid cracking in the layer due to temperature stress, and then repetition of the entire process to add additional layers if necessary. Finally, the fabrication is completed by etching the microfluidic structures in the resin layer with a suitable solvent (propylene glycol methyl ether acetate), thus completing the definition of the master mold. Similarly, and for specific applications, microfluidic structures can be fabricated directly using the SU-8 photoresist as the building material, following equivalent fabrication schemes [79]. In any case, due to the uneven distribution of the photoresist during the centrifugal coating step, the manufacturing process can lead to up to 10% thickness variation in the microfluidic structures. Given the special requirements of SU-8 technology and the long lead times for mold making, lower cost processes based on the use of dry film photoresist have been proposed for mold making without the need for clean room facilities or hazardous chemicals [80]. These dry film-based processes (Figure 3A route 1b) are less expensive and up to 10 times faster than the standard SU-8 technology and, in addition, allow more precise control of the thickness of the structures [79].

From this master mold, several types of materials can be used to replicate the microfluidic chips. Polydimethylsiloxane (PDMS) [42,49,53,73,74,75,76,77] is one of the most widely used materials for microchip fabrication by the classical casting technique [81]. Once cast, it is thermally cross-linked on the microfluidic master mold. Subsequently, the structures are peeled off and bonded to a substrate (e.g., a glass slide) to seal the microfluidic channels (Figure 3B). As an alternative to PDMS for the fabrication of casting chips, OSTEMER [82], a thiol-alkene-epoxy copolymer, should be mentioned. OSTEMER is optically transparent, less porous than PDMS, less permeable, and has high chemical resistance. This material has two independent curing steps, allowing a first cross-linking of the liquid mixture by means of UV exposure, which leads to a solid but flexible material that can be bonded to almost any type of surface by means of the remaining free epoxy groups, which are then cross-linked by a thermal process (Figure 3C).

As an alternative, some optically transparent, thermoplastic materials have also been used for chip fabrication by different manufacturing techniques [83]. Among these techniques, hot embossing (Figure 3D), which also requires a master mold for fabrication, is probably the most widely used to date [84], but there are also techniques allowing direct fabrication (and thus accelerating the step from initial design to prototyping), such as laser engraving and micro-milling [75,85], spark erosion wire cutting [42,48,76], and new additive fabrication processes applied to 3D printing technologies [86]: stereolithography, laminate fabrication and, more recently, melt deposition modeling [87].

In the last five years, paper-based microfluidic devices have gained in popularity [46,88,89,90,91]. As for PMMA, micromachining and laser printing are often used to design micro patterns and structures on paper-type materials [46,90]. Casting wax is generally used during those steps to harden and form the micropattern of such devices [88,89,91].

Gold is commonly used as electrode material in microfluidic BESs as it is conductive and compatible with most of the microfabrication process used in microfluidics [39], even though bare gold could possibly have an altering effect on the redox proteins of the bacterial membrane implied in EET [92]. More typical materials basically used in macro BESs, such as carbon base electrodes, are essentially used in microfluidic BESs with reactors having above-milliliter volumes [39,40] but their minimum sizes are not compatible with the process used to design microliter scale systems [38,41,74]. However, during the last decade, several attempts have been made to use such materials; as in the works of Qian et al. with carbon cloth [72], Nath et al. with carbon nanotubes [46], or Ren et al. with a film of polymer-CNT [92]. Lately, other, metallic materials, such as a Ni-based electrodes have been tested as electrode material in order to improve the microfluidic set-up [75].

### 2.3. The Different Types of BES

Microfluidic microbial BESs can be separated into two categories based on the number of electrodes present inside the set-up. In the first category, microfluidic BESs with two electrodes are based on the operation of MFCs. These microfluidic BESs are either scaled down batch MFCs [38,39,40,41,42,43] or co-laminar flow BESs designed to address the transport limitations of scaled down MFCs [32,44,45,46,47,48,49]. A second category of microfluidic BESs are set-ups with three electrodes, which focus on a more analytical aim, with the incorporation of an additional reference electrode [50,51,52,53].

#### 2.3.1. Scaled Down MFC: Membrane Microfluidic MFC

Membrane microfluidic MFCs are MFCs designed with a reactor volume in the 7 mL to 1.5 µL range [38,39,41,46,73,93,94]. They are basically regular microscale MFCs in which the volume of the reactor is easily lowered from liters to several dozen milliliters to a few microliters in the smallest system [41]. They function in the same way as macroscale MFCs. In a typical MFC, the anode is inoculated with electroactive bacteria (Figure 4A). The bacteria oxidize organic substrates in the electrolyte (anolyte) and then provide electrons to the anode surface, either directly or indirectly. The electrons are then transferred to the cathode through an external resistance, where oxidants (oxygen or ferricyanide are major examples) are electrochemically reduced by an abiotically catalyzed reaction. A proton exchange membrane (PEM) or, more broadly, an ionic exchange membrane (IEM) separates the two chambers to prevent mixing of their electrolytes, while ions including protons can still be exchanged between them. In a membrane microfluidic MFC, the organization is basically the same, except that the electrodes are generally produced directly with microfabrication techniques in order to match the scale of the micro reactor system (Figure 4A).

Initially, a milliliter scale microfluidic membrane MFC and a microliter scale one were differentiated by the fact that the first generated considerable power density whereas the second was shown to provide a better current density but a low power density [63]. For example, a power density up to 10 mW m^−2^ and a current density of 32 mA m^−2^ were obtained with a graphite felt electrode in the work of Ringeisen et al. [39] with a monoculture of *S. oneidensis* MR-1 in a 1.2 mL microfluidic MFC. The authors considered that their milliliter scale MFC produced 160 times more current density and 1960 times more power density than a regular MFC when the real electrode area surface used was taken in account. Microliter scale membrane MFCs [41] produced power densities of up to 1.5 mW m^−2^ and current densities of 130 mA m^−2^ at that time with the same species and a gold electrode.

Nonetheless, microliter scale membrane microfluidic MFCs have been the object of much research during the last decade. Performances have risen to a maximum recorded power density of 830 mW m^−2^ and a current density of 2.59 A m^−2^ when using a mix of polymer and carbon nanotubes as electrode, and inoculating a bacterial consortium enriched with *Geobacter* species [93] in the anodic chamber of 12.5 µL. In contrast, a work with *Shewanella oneidensis* MR-1 in a milliliter scale membrane microfluidic MFC, presenting one of the highest performances to date, displayed a power density of around 661 mW m^−2^ and a current density of 0.4 A m^−2^ with a 3D graphene nickel foam anode [94].

Miniaturizing MFCs to the micro scale is thus far more efficient, as the main advantage of scaling down MFCs is to improve the surface area-to-volume (SAV) ratio. The smaller the reactor volume, the higher the SAV ratio. Improving the SAV ratio has an impact on several aspects of the BES system. Firstly, the ratio between the quantity of bacterial cells adhering to the electrode and the quantity of planktonic bacterial cells tilts more in favor of sessile bacterial cells, which leads to a decrease in interfering biochemical reactions and competition for the biological oxidation of the pool of organic substrates. Secondly, the electrolyte flow at the electrode surface can be modulated upwards since mass transfer limits the overall reaction rate, i.e., as measured by current density. Dominguez-Benetton et al. have shown that the mass transfer coefficient increases when the characteristic electrode length decreases [94]. Assuming that a microscale membrane MFC differs from a macroscale one only by the length of its electrode, miniaturizing the system would forcibly improve the surface maximum flux. Finally, miniaturizing MFC to microscale also leads to a reduction in the internal resistances, by shortening the inter-electrode distance [9,65,67,68,69,70,72,73,95,96,97] or increasing the IEM surface area relative to the smaller size of the electrodes [98]. Microfluidic membrane MFCs are, however, actually considered as low energy output systems, classified among the low power density production systems, because of the presence of the membrane inside the MFC set-up [64].

Oxygen penetration problems related to the gas permeability of the materials used in the microfabrication processes is critical, especially with PDMS, which is entirely permeable to oxygen. Alternative ways to address this issue are to coat the PDMS [74] or to use anodic facultative anaerobic bacterial strains or consortia, less affected by the presence of oxygen.

#### 2.3.2. Membraneless Microfluidic MFC: The Co-Laminar Microfluidic MFC

Taking advantage of the short characteristic length of the microreactor and the high SAV ratio, the fluidic Reynolds number (*Re*), which is the ratio of the inertial forces to viscous forces, is low (less than 100) [99]. When the Reynolds number is low, the flow is laminar rather than turbulent [100]. The mass transport of soluble compounds and ions is then determined by phenomena of diffusion rather than convection. In a co-laminar microfluidic MFC, as two parallel inlets would allow the cathodic and the anodic electrolytes to be injected separately, the fluid flow dynamics would prevent the two electrolytes from mixing (Figure 4B). The implementation of a physical separator such as an IEM membrane (i.e., as in the membrane microfluidic MFC) then becomes irrelevant [101]. Using this idea, Li et al. developed the first membraneless microfluidic MFC in 2011 [49], reaching current densities of 25.4 mA m^−2^ and 18.4 mA m^−2^ with *S. oneidensis* MR-1 and *G. sulfurreducens* electroactive model bacterial strains, respectively, on a gold anode.

Since then, different models of membraneless co-laminar MFCs have been designed with different types of configurations, which can be succinctly divided into Y/T shaped co-laminar MFCs [48,49,77] and membraneless MFCs with exotic geometries [47,75]. The highest performing system reported to date, with carbon cloth electrodes, generated a power density of 3200 W m^−3^, i.e., 1.18 W m^−2^ when the dimension of the limiting electrode was considered, and a current density of 35.5 A m^−2^ [48]. These performances are 1.3 (power density) and 14 times (current density) higher than in the most efficient microfluidic MFC [93].

Like the microfluidic membrane MFCs, co-laminar microfluidic MFCs take advantage of a short response time and a high SAV ratio. In addition, they present a simplified structure, as they do not need a separator membrane. Therefore, they have a lower internal resistance [64] than the membrane microfluidic MFC. The absence of membrane allows the faster transfer of ionic charges in the electrolytes. The precisely controlled laminar flow helps to enhance the system as the stable shear force generated sweeps away weakly adhered cells and favors the formation of a robust anodic biofilm [102]. At the same time, the controlled laminar flow helps to preserve stable concentrations of nutrients in the diffusion layer around the biofilm and participates in a better evacuation of the anodic reaction products, especially protons [103].

Even though this co-laminar microfluidic system presents many advantages, it has some limitations. Due to the laminar flow, which authorizes the transfer of mass through diffusion only, a boundary layer effect, where bacteria accumulate at the entrance of the microfluidic channel, is often observed. The biofilm develops more at the entrance of the channel than it does farther along the electrode, resulting in a thicker biofilm at the entrance and a thinner one at the end. To counter this effect, special architectures with multiple anolyte inlets have been proposed by Yang et al. [33]. A second issue for membraneless microfluidic BESs is the shear rate induced by the flow. A low flow rate would induce a low shear rate, so the boundary layer discussed previously would be thicker and supplying the biofilm with nutrients would be slower, resulting in less effective formation of the biofilm [102]. Conversely, if the flow rate were too high, the high shear stress would alter the biofilm formation on the anode. Partial or total detachment of the electroactive biofilm accompanied by a considerable loss of power density [76] could also occur due to hydrodynamic instability.

#### 2.3.3. Three-Electrode Microfluidic BES

Three-electrode microfluidic BES set-ups are characterized by the presence of a reference electrode to normalize the results recorded with the working electrode (Figure 4C). Such a system allows reliable information to be collected and electrochemical reaction kinetics to be recorded independently on anodes and cathodes. To date, only a few types of three-electrode systems have been elaborated in microfluidic BESs [50,51,53].

One of the solutions proposed to obtain a valid electrode reference is the use of a gold pseudo-reference. The potential versus standard hydrogen electrode (SHE) of the pseudo reference is calculated by evaluating the shift between the redox peak of ferricyanide recorded in voltammetry with the gold pseudo reference and a commercial Ag/AgCl reference in the same bacterial growth media [51,53].With such a set-up, a study by Zarabadi et al. has shown that the bio resistance and bio capacitance of an electroactive biofilm, measured by impedance, is greatly modified by the shear rate applied to it [53]. Another study, using a similar set-up, has demonstrated that increasing the flow rate could mitigate the acidification of *G. sulfurreducens* biofilm but only in starvation conditions [51]. A solution proposed to obtain a reference electrode usable in a microfluidic BES set-up with three electrodes consists of designing a particular architecture with co-laminar fluid specifically selected for the reference electrode. In this system, an Ag/AgCl electrode is fabricated by electroplating Ag on a gold substrate and then oxidizing it in saturated KCl solution. This integrated reference electrode would then be circulated with a saturated KCl solution as the co-laminar fluid. As the mass transfer is assured only by diffusion and not by convection in this co-laminar system, the Ag/AgCl reference electrode would be calibrated and stable for this concentration of KCl [50]. Such a set-up was proven to be stable for over 30 days and demonstrated the precise correlation between the electroactive behavior of *G. sulfurreducens* and the presence of two substrates: ferric citrate and formaldehyde [50].

A major limitation of the three-electrode micro set-up is the necessity to possess a stable reference microelectrode. The Ag/AgCl pseudo references [104,105,106] often used in microfluidic analytical electrochemical cells generally have too short a life span, no more than a few days, when used in a microfluidic three-electrode BES. As it is difficult to design functional pseudo references that form an integral part of the system, some set-ups use commercial references at the electrolyte outlet to characterize their system [52]. In theory, this option is strictly limited to highly conductive electrolytes, i.e., electrolytes highly charged with ions.

Considering the situation described in this overview, the design of easy to fabricate integrable microreferences would be a major advance in this field.

## 3. Contribution of Microfluidic Investigations to the Fundamental Knowledge of Electroactive Biofilms

Fluid flow control and working in microchannels offer the advantages of temporal and spatial control of the physical–chemical conditions of the electrolytes, application of hydrodynamics, and combination with real-time analytical techniques of spectroscopy, microscopy, and electrochemistry to study the many characteristics (Figure 5) that profoundly affect the formation and electrochemical performance of electroactive biofilms. The ability to precisely control and monitor the flow rate in the microfluidic set-up makes it possible to accurately adapt the shear rate applied to the biofilm as well as the concentration of nutrients and substrates. The micro construction techniques also enable exotic dedicated architectures of microchannels to be elaborated and different types of micro sized electrodes to be used to investigate bacterial cell adhesion and EET mechanisms. Inherent in microfluidic systems, such abilities have helped to disentangle some fundamental mechanisms that are particular to electroactive biofilms. They are discussed in the following sections.

Most of the local characterizations aimed at elucidating the relationships between the electronic transfer mechanisms in biofilm and the physical structuring of biofilm on the electrodes have been carried out ex situ by removing the microbial anodes for microscopic imaging or spectroscopic analytical methods. A microbial microfluidic BES, because of its small size and the transparency of its materials (transparent polymers, glass, transparent electrodes), offers a resolutely innovative solution for performing coupled analyses of optical microscopy, spectrometry or electroanalysis (impedance, potentiometry, voltammetry) in situ, in real time and non-invasively.

### 3.1. Contributions to Disentangle the EET Mechanisms

Direct or indirect bacterial EET mechanisms have been widely studied [107,108] and microfluidic tools have aided the investigation of such mechanisms during the last 15 years. Basically, macrosystems allow EET mechanisms to be extensively studied with a wide range of exploration strategies from electroanalytic methods to microscopy investigations and genetic engineering [52]. However, in experimental or analytical macrosystems, it is difficult to be certain whether the result of the observation is attributable to the mechanism itself, or to an assembly of several mechanisms or combined phenomena, or to an artefact due to the myriad of chemical molecules and bacterial species present in the macrosystem. Microfluidics provides significant help in this respect, by allowing more localized interrogation of the system under investigation, by limiting interference reactions, and by applying precisely controlled conditions to the bioelectrocatalytic interfaces. Benefiting from these advantages, Ringeisen et al. proved that microfluidic BESs are proficient tools to discriminate between the direct electron transfer part alone and the total EET driven by *S. oneidensis* [39]. In their work, the current generation obtained in the microfluidic BES was markedly lower, by 30 to 100% depending on the situation, than the current obtained following the addition of an electron shuttle molecule. It is an interesting tool since *S. oneidensis* is known to have an EET resulting essentially from mediated electron transfer [16]. The importance of the mediated EET for *S. oneidensis* is also precisely pictured and studied by combining microfluidic approaches with the implementation of nanoelectrodes in microfluidic BESs. Jiang et al. were among the first to observe that mediated EET was the predominant EET mechanism within *S. oneidensis* [106]. Allying microscopy with microfluidics, they demonstrated that the number of cells attached to the electrode was not directly correlated with current generation, supporting the theory of a main EET based on mediated electron transfer. They also demonstrated that replacing the bacterial growth medium with a new, fresh batch after 48 h would reduce the current by around 95%, which could be recovered to about 80% by switching back to the old medium. This result again strongly supports the predominance of mediated EET for this specific species of electroactive bacteria. Recently, microfluidics has also been used to clarify the mechanistic operation of the EET performed by *Geobacter* nanowires. In their study, Michelson et al. state that nanowires can effectively transfer electrons up to 15 µm from the bacterial cell but they need a bound redox cofactor to work efficiently [60], a result which is consistent with the model theorized by Steidl et al. one year earlier [14]. Another interesting example demonstrating, once again, the contribution of microfluidics to the fundamental field of EET is the discovery of the correlation existing between the polarizability of the bacterial cells and their ability to perform EET. Allying microfluidics and electrophoresis, Wang et al. demonstrate that the faculty to perform EET is directly correlated with the polarizability of the bacterial cells [109]. In their work, the deletion of bacterial genes coding for cytochromes, crucial to the mechanism of direct EET, diminished the polarizability of the genetically engineered cells, whereas the polarizability of cells modified to express a new EET pathway increased. Apart from its high involvement in the rapid screening of electroactive species, this information is of great interest as it could be directly related to relevant mechanisms linked to the formation of electroactive biofilm on electrodes.

### 3.2. Contributions to Our Understanding of Electroactive Biofilm Formation

Biofilm formation is an extensive field of research, which is of great interest beyond the scope of electroactive biofilms, since it is the subject of numerous industrial strategies aimed at increasing or decreasing the adhesion capacity of bacteria. Its relevance to electroactive species is of prime importance, as success in producing a robust electroactive biofilm should allow efficient, sustainable EET. Nevertheless, studies on the impact of the ability to perform EET from electroactive bacterial species during the dynamics of electroactive biofilm formation are scarce. Currently, microfluidic approaches have helped to distinguish several mechanisms that could be involved in this specific topic.

The first mechanism, based on the polarization of the bacterial cell, is not unique to electroactive species but rather is common to all bacteria [109,110]. As the bacteria would aggregate to form a biofilm, the succession of polarization and depolarization of cells synchronizes and leads counter ions such as K^+^, Na^+^, Cl^−^ to accumulate around the biofilm creating a concentration gradient of ions [109]. It could be imagined that the gradient of such ions could drive the planktonic bacterial cells to the biofilm by chemotaxis-like mechanisms. Using microfluidics, Humphries et al. demonstrate that the planktonic bacteria, whatever their species, direct themselves to the biofilm at a rate directly correlated with the oscillation of the polarization of the membranes of bacteria in the biofilm [107]. The formation of the biofilm is then greatly impacted by the electric communication thus generated. This phenomenon is especially interesting in electroactive species, which have high polarizability since they are able to perform EET [111,112].

A second mechanism was demonstrated with *S. oneidensis* based on an energetic chemotaxis called “energy taxy” by the authors [61]. In their work, they successfully showed that bacterial cells moved to follow a gradient of oxidized riboflavin artificially created in a microfluidic BES set-up. In electroactive biofilms, oxidized riboflavin would be self-secreted by the bacterial cells and accumulated around the biofilm as the cell aggregates and the reduced riboflavin would be reoxidized when in contact with the anode. The planktonic cells would then direct themselves to the biofilm by going from one oxidized riboflavin to another, using them as final electron acceptors, following an “energy” gradient.

The last mechanism was based on an electric field gradient. Although this mechanism has not been demonstrated with a microfluidic set-up nor with a millifluidic set-up, the same advantages and properties of microfluidics were used here to carry out the demonstration. Du et al. have pointed out that electroactive bacterial species from wastewater actually respond to an electric field to form a biofilm [110]. Flowing across the electric field, the bacteria from wastewater form a biofilm, centered on the electric field, with a thickness of 88 µm, which is reduced to 30 µm at the edge of the electric field. Metagenomic community investigations performed on this particular biofilm showed that the concentration of *Geobacter* species inside the biofilm was 25% higher at the center of the biofilm than at its edge, implying that, for electroactive species, electric forces could be involved in the formation of the biofilm.

### 3.3. Contributions to Spatial Probing of the Electrochemical Activity of Biofilms

Electroactive biofilms have structural and chemical organizations directly attributable to the bacterial populations from which they originate, their stage of development or maturity, and the local hierarchical organization and co-occurrence of EET mechanisms, all of which depend on the polarization of the electrode on which the biofilm develops [113]. The literature describes patterns of mature biofilms with very different morphologies according to the experimental conditions of electrode potential, the production or absence of polymeric matrix for electron storage [114], monoculture or mixed populations, bioanodes or biocathodes [115,116].

The question has been raised as to whether the electrochemical activity of bacterial cells is homogeneous or not in electroactive biofilms depending on the stage of development of the biofilm, i.e., mainly in connection with its thickness in relation to the electrode surface. The vast majority of investigations have been undertaken with *Geobacter sulfurreducens* model electroactive biofilms, although alternative work conducted with mixed biofilms has shown that the conclusions are not entirely transposable from the single species model system to more realistic multispecies systems [117]. Nevertheless, this *Geobacter sulfurreducens* species is highly represented in mixed anodic biofilms with percentages of representativeness sometimes exceeding 90% [20,118,119]. Blanchet et al. have even demonstrated a strong direct correlation between the maximum current density provided by bioanodes and the relative abundance of *Geobacteriaceae* [118,120]

*Geobacter sulfurreducens* is, therefore, a model of anodic electroactive bacteria capable of forming biofilms with a homogeneous structural appearance and a homogeneous thickness that can exceed 100 µm. The biofilms of *Geobacter sulfurreducens* are conductive and express two coexisting types of direct EET mechanisms: conductive pili and c-type cytochromes. Gene expression of these mechanisms has been reported to be homogeneous throughout the thickness of *Geobacter sulfurreducens* biofilms. This observation was made in real time and in situ, in a transparent chamber equipped for confocal laser scanning microscopy. The fluorescence signal monitored was based on a genetic construction allowing the expression of the reporter gene coding for a short half-life fluorescent protein [121]. Despite this uniform expression of the direct mechanisms of EET, several groups of authors have nevertheless demonstrated that the electrochemical activity of *Geobacter sulfurreducens* biofilms is limited when the biofilms exceed a thickness of 10 µm [14,122,123], i.e., a stratification of about 20 layers of bacterial cells.

The hypotheses put forward to explain why these *Geobacter sulfurreducens* biofilms fail to produce current densities proportional to the biofilm thickness or its biovolume are:
-- Chemical gradients in protons [124,125] or substrates [126] that directly impact cell physiology, viability, and metabolic and respiratory activity. The basal zone closest to the anode is, in principle, the most impacted zone.-- Redox potential gradients that condition the redox state of the c-type cytochrome, with a higher proportion of reduced cytochromes in the layers more distant from the anode surface [127,128,129,130].-- A stratification of the coexisting direct mechanisms of EET: pili become the main electron discharge mechanism in the upper region far from the anode (>10 μm), where the c-type cytochromes are mainly in the reduced state [14].

These explanations and validated experimental demonstrations are often contradictory. For example, some studies report increased cell mortality in the proximity of the anode, while others claim that the biofilm zone in contact with the bulk electrolyte is the zone most affected by mortality. Questions arise as to the possibility of methodological bias or bias related to the sampling and post-treatment of samples before their observation by confocal laser microscopy scanning. To discriminate between these models, the metabolic activity in *Geobacter sulfurreducens* biofilms was measured in a z-profile from stable isotope probes incorporated by bacterial cells, and identified and visualized by coupling with nanoscale secondary ion mass spectrometry (nanoSIMS) [131]. In mature biofilms up to 80 µm thick, this approach concluded that the metabolic activity of bacterial cells follows a gradient, with the most active cells being found on the surface of the anode, and this activity progressively decreases layer by layer with the distance from the surface of the graphite anode.

All these works performed with *Geobacter sulfurreducens* biofilms finally challenge certain theories concerning electronic conductivity in electroactive biofilms, the syntrophy existing between electroactive and non-electroactive populations (direct interspecies electron transfer), and, more generally, the homogeneity of the production of electron flow within the three-dimensional structure of biofilms.

### 3.4. Contributions Examining the Impact of Electrolyte Conditions

The physics and chemistry present inside the electrolyte(s) are key factors conditioning the establishment, performance, and durability of the various pathways of EET performed by the bacteria inside the BES. Oxygen concentration, the nature and concentrations of substrates and nutrients, pH, temperature, buffer strength and electrolyte flow rate (see full details in Figure 2) are crucial factors to be taken into account to achieve an effective BES. Through the advantages it offers for the design of particular BES microreactors or microchannels and for adapting the flow rate, creating gradient concentrations and easily substituting or changing soluble chemical compounds in the extracellular electrolytic medium, microfluidics helps to explore and increase the spectrum of molecules and conditions favorable to electroactive biofilm development and activity.

Using a microfluidics approach, the impact of oxygen on the electrochemical activity of *S. oneidensis* was observed, providing evidence of a change in metabolism in the presence of oxygen, and a preference to use oxygen rather than an anode as a terminal electron acceptor [61]. On the other hand, the culture of *S. oneidensis* in a microfluidic BES under aerobic conditions revealed a more diversified use of the number of substrates that the bacterial strain was able to oxidize to transfer electrons to the anode. In the microfluidic MFC they designed, Biffinger et al. demonstrated that glucose and fructose were serious candidates for use by *S. oneidensis* in aerobic conditions [130]. In this case, microfluidics significantly improved the precision of the results thanks to the scaling down of the system, and the fluidic approach enabled accurate control of the flow in the experiment. These advantages are especially handy as the efficiency of EET depends on the substrates and the condition of the electrolyte.

The effect of pH upon the EET has also been well studied, especially for *Geobacter* species [44,51,122]. Franks et al., by associating real-time imaging and microfluidic management, monitored the effect of the accumulation of protons upon a biofilm formed by *G. sulfurreducens* [122]. The accumulation of protons led to EET inhibition and therefore decreased anodic current generation. This inhibition was reversible, since further pH management to maintain a neutral pH brought the anode current density back to its maximum level. In a recent work, Zarabadi et al. attempted to solve this pH inhibition problem by increasing the flow rate in their microfluidic BES with the aim of promoting the release of protons out of the biofilm and consequently limiting the acidification [51]. Unfortunately, the strategy was only effective with a flow rate that simultaneously led to a starvation mode for the bacteria inside the biofilm.

The importance of flow dynamics and its consequences on the formation of biofilms on solid surfaces in microfluidic systems has been really well studied with non-electroactive bacteria [102,132,133], whereas the same type of studies on biofilms of electroactive bacterial species are limited [51,103,134]. In particular, microfluidic work with non-electroactive bacteria has shown that the shear force applied by the circulating fluid has many effects on a bacterial biofilm, from the early stages of biofilm formation, where it traps planktonic cells on a solid support, to its inhibition by high shear forces causing erosion of the biofilm [102,133]. Microflow dynamics can also affect the mass transport of soluble molecules, particularly nutrients, molecules of quorum sensing and protons, thus playing a major role in maintaining the condition of the extracellular medium, i.e., the electrolyte for the microfluidic BES. The flow rate is then crucial as, if it is too low or too high, it can negatively influence the growth of a sturdy and efficient biofilm. In the intention of estimating the best flow rate for electroactive species, Vigolo et al. defined an optimal flow condition of 20 µL. min^−1^ with *S. oneidensis* [101]. However, it is important to understand that this optimal flow rate could be dependent on the anode chamber volume, the bacterial species constituting the biofilm, and the EET pathway expected in the system, i.e., whether it is essentially based on direct or mediated EET. In another work, Ren et al. also underlined the importance, for the bacteria attached to electrodes [132], of mass transfer through hydrodynamic management, since it would be important to recharge the cytochromes c excreted on the biofilm matrix, as was predicted by an enzymatic model with cytochromes c alone [135].

A microfluidic set-up also informs on the importance of the electrode chamber configuration. Luo et al. showed that better formation of biofilm and higher current generation were obtained when the hydraulic retention time (HRT) of the electrode chamber was increased [45]. However, a balance between this parameter and the internal resistance of the microfluidic BES is needed as the rise of HRT values is strongly correlated with an increase in the internal resistance. On this topic, Choi and Chae demonstrated that a minimum height of 20 µm was absolutely necessary in the electrode chamber to allow the formation of a multilayer biofilm in a microfluidic BES [136].

## 4. Outlook

Over the last decade, microfluidic tools have significantly helped to clarify the fundamental domain of bacterial EET, as well as proposing new optimization strategies to improve and design new BES systems, and microfluidics will no doubt continue to contribute to several exciting strategic areas of research. When thinking about what microfluidics can do to help the study and progress of the BES field, it is important to focus on the strengths that microfluidics systems possess. Microfluidic BES systems are characterized by their ability to precisely control the flow, to manipulate (sort, fuse, aggregate) bacteria, cells, droplets and solid particles, and the possibilities they offer for building particular architectures of experimental systems that are transparent and thus compatible with many real-time macroscopic and microscopic in situ observation techniques, and other non-destructive analytical strategies by spectroscopy. There are many transparent electrode materials commercially available as they were originally developed for solar cell technologies. Some of them, such as indium tin oxide (ITO), modified ITO with transparent deposits of graphene, graphene oxides or PEDOT, or fluorine-doped ITO (FTO), have already been used as a support for electroactive biofilms. ITO has been the subject of several studies with the electroactive bacterial strain *Geobacter sulfurreducens* [127]. FTO, which has a higher electrical conductivity, close to that of carbon, has recently made it possible to visualize the macroscopic development of an electroactive biofilm formed from wastewater in real time and thus acquire kinetic data on biofilm formation [137].

The very attractive feature of microfluidics is the possibility it provides to precisely and almost instantaneously control the flow inside a process and thus, in the specific case of BES, at the interface between the electroactive biofilm and the electrolyte. This property, specific to microfluidics, makes it possible to work in greater depth on many important factors interacting with the EET performed by the bacteria with the electrode, such as the nutrient concentration, the pH, the concentration of oxygen or any molecule, enabling us to create gradients or to quickly switch the flow rate of the circulating electrolyte [138]. Then, microfluidics also allows the importance of shear force for the electroactive biofilm and its electroactivity [103,134] to be characterized. In addition, it offers a microfluidic BES a quick response to any modification of its electrolyte properties [63,139,140]. All these attributes are already well exploited in microfluidic BESs. However, much more promising research is waiting to be done, as most fundamental results have so far been obtained with monoculture biofilms, i.e., biofilms formed from model single electroactive bacterial strains. The diversity of the bacterial community present in biofilms established from complex natural or industrial media, such as wastewater sludge or digester sludge or marine soils, presents a future field of investigation that will also be interesting for microfluidic BESs. The simplification and precise control inherent in microfluidic approaches could help unravel the participants and mechanisms involved, the mechanistic and partnership interactions within bacterial communities, and the pattern of biochemical reactions and electronic exchanges that occur at the scale of an electroactive biofilm.

This ability to precisely control the flow at the precise level of the interfaces of interest, added to the high adaptability of the experimental systems in terms of materials and architecture, sets microfluidics apart as a platform available in all contexts and limited only by the usage the experimenter requires. The possibility of manufacturing simple, microscope-adaptable microfluidic devices for real-time observation [141], where operational parameters can be easily modulated [142], opens the door for in situ investigation of the early stages of bacterial adhesion [143] and the formation of electroactive biofilms directly at the microscale [144]. The compatibility of most microfluidics materials with observational methods allows for easy monitoring and probing of the electroactive biofilm in real time—a real asset when the transient nature of biofilm is considered. A wide range of microscopic techniques, from confocal microscopy [145] and scanning electrochemical microscopy [146] to spectroscopic techniques [139,147,148], have been successfully coupled into microfluidic platforms for live observation of biofilm. Other techniques can also be integrated, such as the widely studied optical density for quorum sensing analysis [149,150] or the innovative use of electrochemical imaging, where specific positions of a voltammogram are converted into pixels, enabling identification of redox currents and peak positions of an electroactive biofilm [151]. The combination of microbial engineering approaches, including matter balance transformation, target microbe selection, mutant characterization, and microbial function analysis, with microfluidic BESs will provide really new information as is already a trend in other domains exploring the activity and behavior of microorganisms in microfluidic study systems [140]. Considering the strengths of microfluidic BESs in screening electroactive bacterial strains [55], the association of the two strategies would lead us to discover and investigate new EET pathways, as in extremophile species, for example, which are difficult to study in macro systems even though they give fairly convincing electrochemical results [141]. The large amenability and compatibility of microfluidic set-ups also encourages great hopes that this technology will form part of an all-inclusive micro research platform for electroactive biofilm characterization.

## 5. Conclusions

Work on microfluidic BESs has grown and flourished in their short ten-year life span and is having a significant impact on fundamental scientific advances in the field of microbial bioelectrochemistry. The progress made on the integration of polyphasic electrode–electrolyte systems at the micrometer scale associated with the availability of a wide variety of materials means that, today, the architecture, composition and arrangement of electrodes in microfluidic BESs seem to be limited only by the objectives of the scientists and the means available to them. Microfluidic BESs can be succinctly divided into two categories: microfluidic BESs with two electrodes, which include microfluidic membrane MFCs and co-laminar MFCs, or microfluidic membraneless MFCs; and BESs with three-electrode systems, including the addition of a reference electrode. Over the years, microfluidic BESs have contributed to the fundamental field of EET by disentangling some fundamental mechanisms, such as the importance of mediated EET for *S. oneidensis*, or by confirming hypotheses and models about EET predicted from macro systems. Thanks to their advantages, microfluidic approaches have also made it possible to propose multiple hypotheses on how the EET properties of electroactive bacteria could interfere with or participate in the formation of electroactive biofilms on electrodes and modulate the performance of their electroactivity. Finally, microfluidic BESs have proved to be interesting for their huge potential to advance the understanding of EET mechanisms and electroactive bacteria behavior and interaction to a new level, as they are compatible with numerous analysis strategies, such as real-time monitoring or microbial engineering.

## Figures and Tables

**Figure 1 microorganisms-08-01841-f001:**
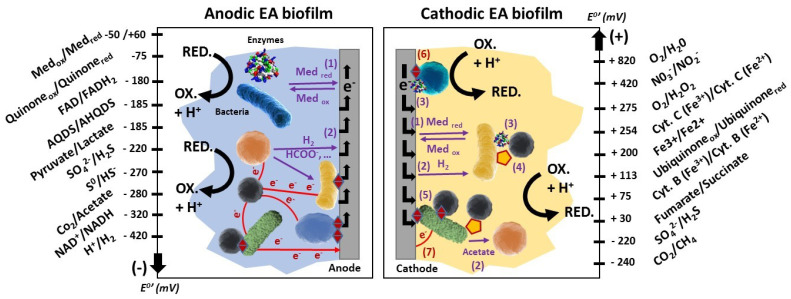
Overview of the bacterial external electron transfer (EET) mechanisms described in the literature and operating individually or in combination in anodic and cathodic biofilms. This figure was adapted from Aghababaie et al. 2015 [8], Santoro et al., 2017 [9], and Blasco-Gomez et al., 2017 [10].

**Figure 2 microorganisms-08-01841-f002:**
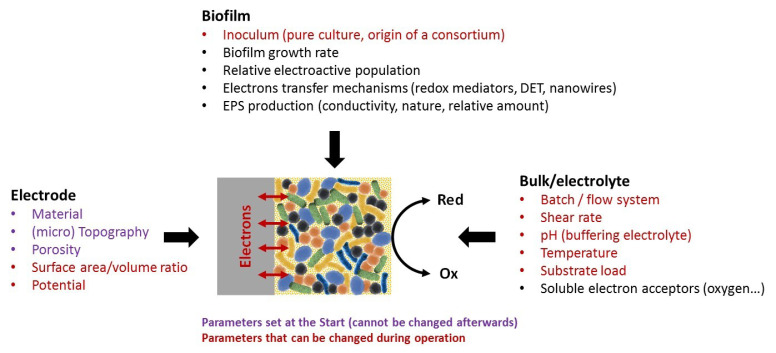
Electrode, biofilm, or electrolyte-related parameters known to have a significant impact on the formation and performance of microbial electrocatalytic systems.

**Figure 3 microorganisms-08-01841-f003:**
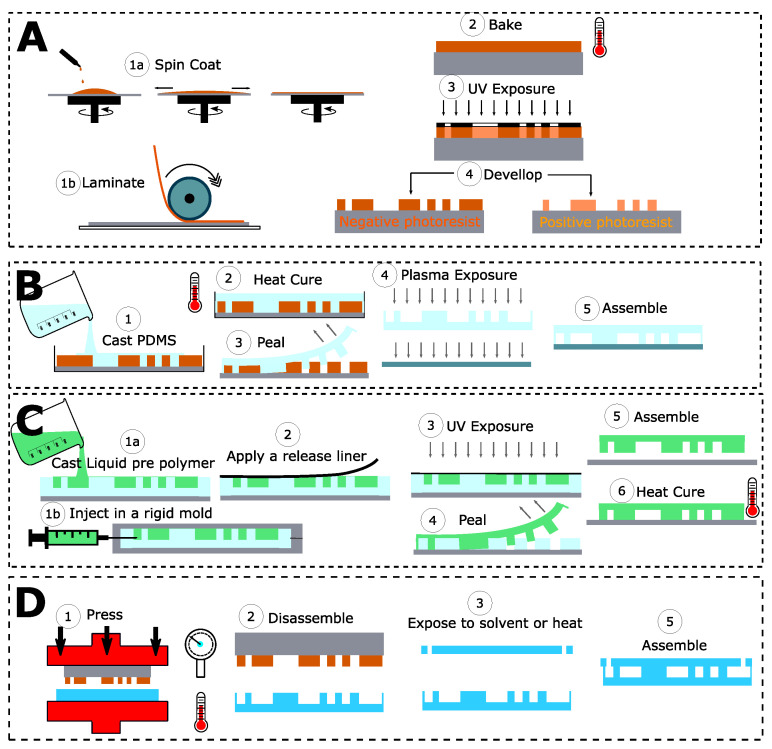
Overview of the most widely used techniques for master-mold and microfluidic chip fabrication. (**A**) Mold fabrication technique using liquid photoresist (1a) or dry film photoresist (1b). (**B**) Microfluidic chip fabrication using polydimethylsiloxane (PDMS). (**C**) Microfluidic chip fabrication using dual cure liquid-prepolymer (Epoxy, OSTEMER, NOA, thiolene, etc.) by cast molding technique (1a) or reactive injection molding (1b). (**D**) Microfluidic chip fabrication using thermoset polymers by hot embossing.

**Figure 4 microorganisms-08-01841-f004:**
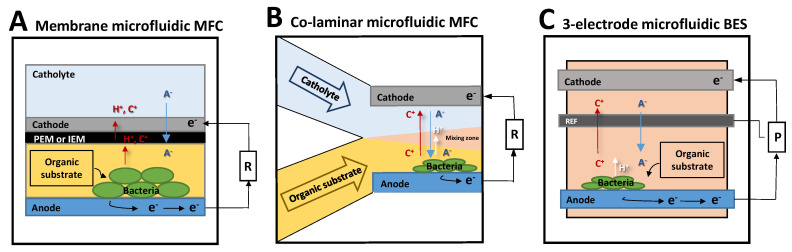
Presentation of the three categories of microfluidic bioelectrochemical system (BES). (**A**) small-scale membrane microbial fuel cell, (**B**) co-laminar electrode microfluidic, (**C**) BES with the incorporation of an additional reference electrode.

**Figure 5 microorganisms-08-01841-f005:**
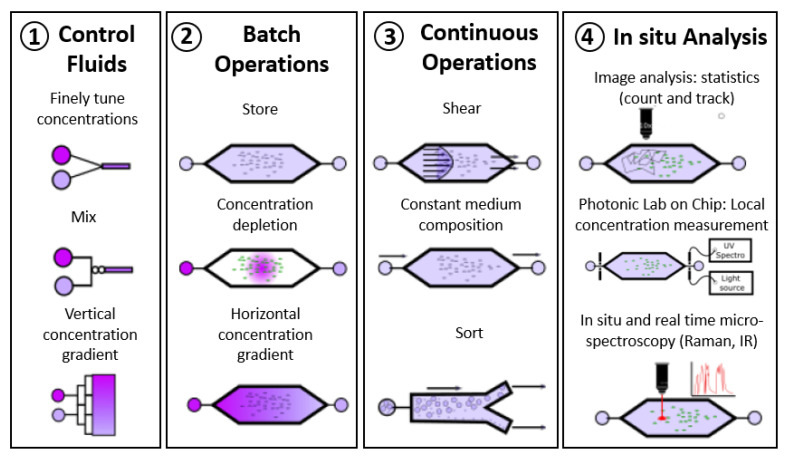
Advantages offered by microfluidic set-up for studying cells. The microfluidic set-up takes advantage of a fine control of hydrodynamics, (**1**) on the way that the microfluidic device is operated (batch (**2**) or continuous mode (**3**)) and on the opportunity to perform local and in-situ analyses (**4**).

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
