# Peer review of "Microfluidic Microbial Bioelectrochemical Systems: An Integrated Investigation Platform for a More Fundamental Understanding of Electroactive Bacterial Biofilms"

_microorganisms, 2020, doi:10.3390/microorganisms8111841_

Round 1

Reviewer 1 Report

The presented work sets out to review the growing field of microfluidic bioelectrochemical systems and how it can be used to gain new insights into electroactive microbial biofilm formation, performance and EET mechanisms. This topic is most interesting to the scientific community and the authors outline that significant advances in fundamental understanding of electroactive bacteria from microBESs will be presented. While the premise is well motived, very few concrete examples of significant advances in understanding of electroactive bacteria based on microBES studies are offered. Instead the authors detail loose concepts of electroactive biofilm growth and its dependence on hydrodynamic and bioelectrochemical factors with little to no scientific discussion of the underlying mechanisms that are targeted with microBESs. 

Morever, the language is not acceptable for any scientific publication as there are multiple misspellings and even factual errors. A few are here listed to exemplify the problem: 

Throughout the manuscript examples are often written as "..." instead of etc. 

This is hardly acceptable for a scientific publication.

Abstract: "monoclonal electroactive bacteria"

As far as I am aware, the term monoclonal is used in terms of antibodies and not to describe bacteria. 

p4 line 156: "Polydymethylmethylsilsyloxane (PDMS)"

PDMS is polydimethylsiloxane, I am not certain which chemical the authors are referring to here. 

p7 line 273: "14 times (current density) more important"

How can you quantify importance? 

I believe that the topic should be adressed but the submitted work does not improve the understanding of the topic and is not in a presentable form fitting for a scientific publication. I must therefore recommend rejection in its current form.

Author Response

Reviewer 1:

The presented work sets out to review the growing field of microfluidic bioelectrochemical systems and how it can be used to gain new insights into electroactive microbial biofilm formation, performance and EET mechanisms. This topic is most interesting to the scientific community and the authors outline that significant advances in fundamental understanding of electroactive bacteria from microBESs will be presented. While the premise is well motived, very few concrete examples of significant advances in understanding of electroactive bacteria based on microBES studies are offered. Instead the authors detail loose concepts of electroactive biofilm growth and its dependence on hydrodynamic and bioelectrochemical factors with little to no scientific discussion of the underlying mechanisms that are targeted with microBESs. 

In the end, not so much work has been done with microBESs that has led to concrete progress in the discovery of mechanisms. Most of the major discoveries around electroactive biofilms have been made with macroBES. We have limited this review article to summarizing the results of the work done with microBES. Nevertheless, following your advice, we have established a new section on "Contributions to spatially probing the electrochemical activity of biofilms" in which more fundamental contributions are deciphered.

Morever, the language is not acceptable for any scientific publication as there are multiple misspellings and even factual errors.

We are sorry for that. The manuscript has been screened to satisfy your recommendations. Also, the manuscript has been taken entirely over by a company specialized in proofreading English documents in order to improve the language quality.

A few are here listed to exemplify the problem: 

Throughout the manuscript examples are often written as "..." instead of etc. 

This is hardly acceptable for a scientific publication.

This is now corrected

Abstract: "monoclonal electroactive bacteria"

As far as I am aware, the term monoclonal is used in terms of antibodies and not to describe bacteria.

You are absolutely right.

Monoclonal was replaced by monoculture.

p4 line 156: "Polydymethylmethylsilsyloxane (PDMS)"

PDMS is polydimethylsiloxane, I am not certain which chemical the authors are referring to here.

Indeed, the error has been corrected

p7 line 273: "14 times (current density) more important"

How can you quantify importance? 

I think this expression is often found in other documents. But as requested by the reviewer, we proceeded to the change (higher than...)

I believe that the topic should be adressed but the submitted work does not improve the understanding of the topic and is not in a presentable form fitting for a scientific publication. I must therefore recommend rejection in its current form.

I can't completely say "thank you" here but your comment at least allowed us to explore a very informative additional point about “spatial probing” now added to the discussion.

Reviewer 2 Report

In the present review, the authors investigate the fundamental and coupled phenomena that occur during the formation and expression of electrocatalytic activity in electroactive biofilms. In particular, they focused their attention on miniaturized microBES, explaining all phenomena occuring at the microscale. In this review article, the configurations of microfluidic BES designed for selected objectives and their microfabrication techniques are detailed. Based on this, the review can be accepted for publication on Micoorganisms. Some references can be added to complete the literature citations and the quality of images should be improved: in particular Figure 1, Figure 4 and Figure 5

Author Response

Reviewer 2:

In the present review, the authors investigate the fundamental and coupled phenomena that occur during the formation and expression of electrocatalytic activity in electroactive biofilms. In particular, they focused their attention on miniaturized microBES, explaining all phenomena occuring at the microscale. In this review article, the configurations of microfluidic BES designed for selected objectives and their microfabrication techniques are detailed. Based on this, the review can be accepted for publication on Micoorganisms. Some references can be added to complete the literature citations and the quality of images should be improved: in particular Figure 1, Figure 4 and Figure 5

Thank you very much for your very positive return

Also in line with the comments of reviewer 3, you can see that strategic references have been added in the manuscript. You will also see that a new discussion has been integrated in section 3.

We don't know if you used a .doc or .pdf file as a support for your review. In the .doc document we provided the resolution of the figures is very good. Figure 5 has a lower resolution than the others. In the revised version of the manuscript we have replaced this figure with a high-resolution figure and also the figure 2.

If the problem persists, we propose to supply the individual figures independently in image format to the journal.

Reviewer 3 Report

General comment: please review the use of the acronym EA and the word electroactive. Uniform the use throughout the text.

General comment: please revise the manuscript editing.

Line 35: Please unify Bio Electrochemical Systems or bioelectrochemical Systems.

Line 54: replace “environmental waste” by “organic waste”.

Line 56: enumerate some of the mentioned electroactive bacteria. Also, in the same paragraph, it would be worth reinforcing that EA bacteria respond differently in electron transfer/uptake terms, according to the electrode potential. Also, interfacial micro condition can affect EA biofilms.

Line 93: please mention the volume of the 3 electrodes set-u, specify if it is in the milliliter or microliter range.

Line 106: stress out which are the mentioned implications.

Line 129: some references are missing.

Line 139: ad reference for SU-8 Technology.

Line 187: If carbon-based materials used in macrosalce BES cannot be generally employed in microfluidic BES, how can the finding in the latter transferred to the first ones? If one aims at studying the EA biofilm-electrode interaction, the tested materials should be those employed at larger scales. Otherwise, the findings are not directly transferable to higher scales.

Section 2.3/Figure 4: Only MFC like microfluidic BES are represented. Othe type of BES reactors, including EA biofilms associated to the cathode are missing.

Line227: reference is placed incorrectly.

Line 236: If SAV ratio is favored at micro scale, how can the findings then be transferred to higher scale? There might be some bias in this point. Authors could bring some insight on this.

Section 3: presents a good overview of the advances that microfluidic BES have allowed in the recent years in understanding fundamental processes in BES. In this section, it is missing information regarding previous works on the use of in-situ observation techniques and analytical strategies in microfluidic BES and how they impacted on basic knowledge in BES operation. I suggest, to complete the section, to look for references using microfluidic systems for studying the electroactive biofilms with omi tools, to gather information on how microbial metabolism is regulated by electrochemical conditions.

Author Response

Reviewer 3: 

General comment: please review the use of the acronym EA and the word electroactive. Uniform the use throughout the text.

Yes, it is now fixed

General comment: please revise the manuscript editing.

Line 35: Please unify Bio Electrochemical Systems or bioelectrochemical Systems.

Done

Line 54: replace “environmental waste” by “organic waste”.

Done

Line 56: enumerate some of the mentioned electroactive bacteria.

The 2 lists (microbial anode and microbial cathode) have been completed

Also, in the same paragraph, it would be worth reinforcing that EA bacteria respond differently in electron transfer/uptake terms, according to the electrode potential. Also, interfacial micro condition can affect EA biofilms.

Please see lines 70 to 77.

Line 93: please mention the volume of the 3 electrodes set-u, specify if it is in the milliliter or microliter range.

Done

The volume was not reported but we added the following information “where at least one of the dimensions (height, length, or width) is micrometric”

Line 106: stress out which are the mentioned implications.

Done. See Lines 113-115.

Line 129: some references are missing.

Done. References 48, 49, 50 and 52 were added.

Line 139: ad reference for SU-8 Technology.

Done. The reference “SU-8 as a structural material for labs-on-chips and microelectromechanical systems” was added

Line 187: If carbon-based materials used in macrosalce BES cannot be generally employed in microfluidic BES, how can the finding in the latter transferred to the first ones? If one aims at studying the EA biofilm-electrode interaction, the tested materials should be those employed at larger scales. Otherwise, the findings are not directly transferable to higher scales.

Yes, of course, it is quite relevant to ask this type of question. As a comment, it can be argued that work has already identified that mixed electraoctive biofilm communities are very similar on very different electrode materials, platinum vs. carbon for example, or stainless steel and carbon. On the other hand, model work with Geobacter has shown macroscopic behaviors very similar to the biofilms of this strain on gold and steel microelectrodes. So yes indeed there are opportunities to explore small-scale mechanisms in "microsystem" devices. But first of all, we must be careful to miniaturize the experimental systems while being consistent with what happens in macroscopic experimental systems. The design of cells and experimental procedures is therefore a primordial issue.

Section 2.3/Figure 4: Only MFC like microfluidic BES are represented. Othe type of BES reactors, including EA biofilms associated to the cathode are missing.

The review focuses mostly on bioanodic systems. Figure 4.C does not represent a schematic of MFC but of an electroanalytical system with three electrodes connected to a potentiostat. This experimental design can be used to study both anodic and cathodic biofilms.

Line227: reference is placed incorrectly.

Now corrected.

Line 236: If SAV ratio is favored at micro scale, how can the findings then be transferred to higher scale? There might be some bias in this point. Authors could bring some insight on this.

There is no issue here of transposing the SAV ratio from a MACRO system to a MICRO system and vice versa. The explanation is, it seems to us, already argued in the current text. The change of scale which is accompanied by an increase in the SAV ratio makes it possible to focus the study on a particular phenomenon without being limited by other phenomena. It is really a concern of analysis that it is necessary to see, and not to be able to go back and forward between micro and macro considerations.

Section 3: presents a good overview of the advances that microfluidic BES have allowed in the recent years in understanding fundamental processes in BES. In this section, it is missing information regarding previous works on the use of in-situ observation techniques and analytical strategies in microfluidic BES and how they impacted on basic knowledge in BES operation. I suggest, to complete the section, to look for references using microfluidic systems for studying the electroactive biofilms with omi tools, to gather information on how microbial metabolism is regulated by electrochemical conditions.

Thank you for this remark which has been very seriously taken into account. In response we have written a new section entitled : 3.3 Contributions to spatially probing the electrochemical activity of biofilms. I hope that you will find through this new discussion the elements you think were missing.

Round 2

Reviewer 1 Report

The authors have made significant improvements to the manuscript and I can therefore gladly recommend it for publication after correction of a few minor issues, as detailed below.

Minor corrections still required: 

The quality of the figures varies greatly. Fig 1, 2 and 5 show a mix of pixelated poor quality graphics with high resolution text. While Fig 3 and 4 are presented in high resolution vector based graphics. Please update all figures so that they are equally presented in high resolution of at least 300 dpi. 

Figure 1 is very difficult to process. There is a lot of information that the authors want to present and correlate between bioanodes and biocathodes. But it is hardly readable. I suggest reworking the figure and separating the different EET mechanisms spatially. Also update to vector-based graphics for better resolution.

p11, l425: "..." is still used instead of etc. Please correct this and verify that the mistake is not present elsewhere in the updated manuscript. 

Author Response

As asked by the reviewer 1, the figures 1, 2 and 5 have been changed in format. I hope that the new resolution will make them more visible. It seems to me that the quality is now satisfactory.

Thank you